# SEDiT: Mask-Free Video Subtitle Erasure via Diffusion Transformer

## Abstract

Recent breakthroughs in video diffusion models have significantly accelerated the development of video editing techniques. However, existing methods often rely on inpainting video frames based on masked input, which requires extracting the target video mask in advance. The precision of the segmentation directly affects the quality of the completion. In this paper, we present **SEDiT**, a novel one-stage video **S**ubtitle **E**rasure approach via **Di**ffusion **T**ransformer. We introduce a mask-free inference approach, which enables direct erasure of targeted subtitle. The proposed one-stage framework mitigates the suboptimality inherent in the two-stage processing of prior models. To address the challenge of long-term temporal consistency, we adopt a hybrid training strategy by occasionally conditioning the model with clean first-frame latent. This facilitates temporal continuity, allowing each segment during inference to leverage the output of its predecessor. To avoid visible seams caused by cropping and reinserting processed targets, particularly in scenarios involving substantial motion, we feed the original video directly into SEDiT. Thanks to the highly compressed Variational Autoencoder (VAE) in the base model and chunk-wise streaming inference, our method can efficiently handle naive 1080p video with infinite length.

## 1 Introduction

In recent years, video diffusion models Blattmann et al. (2023); Kong et al. (2024); HaCohen et al. (2024); Wan et al. (2025) have made remarkable progress in video generation. Current mainstream models employ the Diffusion Transformer (DiT) Peebles & Xie (2023) architecture to achieve unprecedented generation quality. Video inpainting, a key technique in the field of video editing, has also benefited from advancements in foundational video generation models, leading to a significant improvement in restoration quality.

Video object removal is a technique designed to eliminate designated objects from video sequences while maintaining spatial background consistency and temporal coherence. An early GAN-based representative method, ProPainter Zhou et al. (2023), addresses video inpainting by completing the optical flow within masked regions, combined with a mask-guided sparse Transformer to handle the video inpainting task. Due to limitations in model size and the inherent constraints of GAN-based inpainting methods in generative capability, such approaches tend to produce visual artifacts in complex scenarios. Recently, video inpainting methods based on video diffusion models have gained popularity, with notable examples including DiffuEraser Li et al. (2025), Minimax-Remover Zi et al. (2025), and EraserDiT Liu & Hui (2025). DiffuEraser utilizes the preliminary completion results of a prior model (Propainter (Zhou et al., 2023)) combined with DDIM inversion techniques to obtain better initialization. Minimax-Remover is built on the Wan2.1-1.3B Wan et al. (2025) model as its base model. Replace the original text conditions with learnable contrastive tokens to guide the removal process. These tokens are seamlessly embedded into the self-attention stream, enabling the complete elimination of cross-attention layers from the pre-trained video generation model. Recently, EraserDiT Liu & Hui (2025) has employed a more compact base model, LTX-Video HaCohen et al. (2024), to enable high resolution (1080p) video object erasure. It integrates with the Vision Language Model (VLM) Bai et al. (2025b) to interactively remove specified targets from video content.

Despite rapid advances in video object removal technology, existing methods still face numerous challenges. Some approaches produce unwanted content or artifacts, while others introduce blurring or temporal flickering in the regions to be erased. Moreover, previous methods are heavily relying on precise masks, which in turn depend on the accuracy of video segmentation algorithms. Subtitle removal in video represents a critical sub-task within the broader domain of object removal. Effective subtitle removal in videos requires precise and fine-grained segmentation during the pre-processing stage. However, subtitles often exhibit complex visual properties such as transparency, stylized fonts, and gradient effects, which significantly hinder accurate segmentation. These intricacies significantly impact the overall performance and reliability of subtitle removal systems.

Generic object removal is more suitable for interactive editing scenarios, where the user manually selects the target object to be removed in the first frame, followed by the application of video object segmentation algorithms (e.g., SAM2 (Ravi et al., 2024)) to obtain the corresponding mask. This process typically involves single-shot videos with a duration of approximately 10 seconds. In contrast, subtitle removal operates over the entire video, which often consists of multiple shots. Since the removal target (subtitle) is explicitly defined, the task can be performed without user interaction.

In general, enlarging the removal area expands the inpainting scope, which heightens the risk of artifacts, blurring, and other visual distortions, thereby substantially increasing the challenge of achieving seamless restoration. As illustrated in Figure 1a, most previous methods utilize bounding boxes produced by optical character recognition (OCR) algorithms to encompass the entire subtitle region. However, this approach suffers from significant limitations: it discards crucial pixel-level guidance information embedded within character gaps, which severely restricts the achievable quality of restoration. To obtain fine-grained text masks, it is necessary to train a dedicated text matting model, which adds to the overall complexity of the video subtitle removal system.

To address these limitations, we propose an one-stage framework for video subtitle removal that eliminates the need for OCR to localize subtitles. As illustrated in Figure 1b, we eliminate the OCR module and instead rely solely on the predefined prompt instruction to accomplish subtitle erasure. Inspired by recent advances in instruction-based image editing algorithms such as *FLUX.1 Kontext* Labs et al. (2025) and *Qwen-Image* Wu et al. (2025), we propose *SEDiT*, a method that operates without explicit masks and performs the task using prompt-guided control alone. This simple yet effective design is particularly well suited to video subtitle removal, a task that typically requires no user interaction and benefits from efficient batch processing. To enable direct processing of 1080p video, we adopt LTX-Video HaCohen et al. (2024) as the backbone model, which incorporates a VAE with very high spatial-temporal compression. To fully exploit the capabilities of the base model while minimizing structural modifications, we adopt the same conditioning strategy as *FLUX.1 Kontext*, where the conditional video is concatenated along the sequence dimension. In addition, to obtain paired training data, we designed a comprehensive data synthesis pipeline that includes font attribute configuration, subtitle orientation and positioning, as well as transition effects. This setup covers most subtitle scenarios and closely approximates the conditions found in real-world applications. To accommodate the demands of long-form video subtitle removal, we adopt a chunk-wise processing strategy. Except for the first chunk, each subsequent chunk leverages the last frame of the preceding chunk as a reference to enhance temporal consistency. Due to the strong conditioning capability of the reference video, satisfactory temporal consistency across chunks can be maintained by referencing only a single frame between adjacent segments.

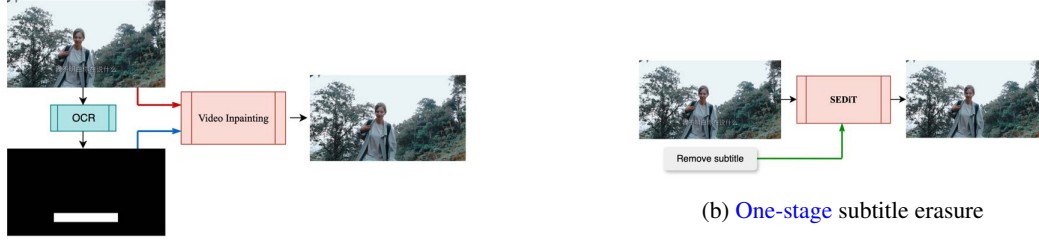

(a) Two-stage subtitle erasure

(b) One-stage subtitle erasure

Figure 1: Frameworks comparison between previous mask-based subtitle removal methods and our mask-free one-stage method.

Our main contributions can be summarized as follows.

- We present SEDiT, a mask-free one-stage framework designed for video subtitle erasure. Unlike previous mask-based video inpainting approaches, our method eliminates the need for explicit mask generation to accomplish subtitle removal. We have developed a relatively comprehensive data synthesis pipeline that incorporates font attributes, subtitle rendering directions, and transition effects. Thanks to the high-quality synthetic data, our method achieves promising results in video subtitle removal.

- To inject the reference video without modifying the model architecture, we concatenate conditional video latent and noisy video latent along the sequence dimension. This extends the prevailing image editing paradigm into the domain of video editing. This approach significantly reduces training complexity while preserving strong reference fidelity. Moreover, this conditioning strategy can be easily applied to other video backbone models, enabling flexible integration of video control signals without structural modifications.

- To enable the processing of video segments of arbitrary length, we adopt a chunk-wise strategy to handle the video. The chunk size is dynamically adjusted based on the resolution of the conditional video, with higher resolutions corresponding to smaller chunk sizes. To mitigate temporal discontinuities between chunks, the last frame of the preceding chunk is used as the first frame of the noisy video latent in the subsequent chunk. Owing to the strong conditioning effect of the reference video, a single-frame condition is sufficient to achieve satisfactory temporal coherence. For tasks involving strong reference-based video editing, this is a simple yet effective strategy.

## 2 RELATED WORK

### 2.1 VIDEO DECAPTIONING

Video Decaptioning amis to remove subtitles from videos and repair the occluded regions. The BVDNet Kim et al. (2019) constructs an encoder-decoder model, which is a blind video decaptioning framework. Chu et al. (2021) proposes a generic video decaptioning framework with two stages. The first stage is a caption mask extraction model. The second stage is a decaptioning model. Both methods mentioned above utilize the ECCV ChaLearn 2018 LAP Video Decaptioning Challenge dataset. Each sample in the dataset is a 5 seconds MP4 video clips, including 125 RGB frames of $128 \times 128$. The resolution of this dataset is too low, making the trained model insufficient for handling subtitle removal tasks on 1080p or higher-resolution videos. To address this, we collected high-resolution, high-quality subtitle-free data during the construction of the training dataset, and paired it with a relatively complete data synthesis pipeline to simulate common subtitle types as accurately as possible. Subsequent experimental results show that our method significantly improves both the effectiveness and generalization of subtitle removal.

### 2.2 IMAGE EDITING VIA PROMPT INSTRUCTION

In the field of image editing, InstructPix2Pix Brooks et al. (2023) has demonstrated the potential of the instruction-guided diffusion model for performing image editing tasks. Recently, *FLUX.1 Kontext* Labs et al. (2025) has achieved significant success in the domain of instruction-guided image editing. It is a simple flow matching model that concatenates context and instruction tokens *along the sequence dimension* and directly estimates a velocity prediction target. Very recently, *Qwen-Image* Wu et al. (2025) has also proposed a similar approach by incorporating the VAE-encoded latent representation of the input image into the image stream, concatenating it with the noisy image latent *along the sequence dimension*. These suggest that, in the field of image editing, concatenating the conditional image latent with the noised image latent along the sequence dimension has become the prevailing paradigm. Inspired by these works, we investigate strategies for incorporating conditional video inputs in the domain of video editing.

### 2.3 DIFFUSION-BASED VIDEO INPAINTING

Diffusion-based video inpainting falls under the category of masked video-to-video editing (MV2V), it requires mask sequence of the region of interest (ROI) $\bar{\mathbf{m}} \in \mathbb{R}^{f \times h \times w \times c_m}$. This approach

completes masked video using a conditional diffusion model $u_\theta\left(\mathbf{z}_t, \mathbf{z}_m, \bar{\mathbf{m}}, \text{prompt}, t\right)$, where $\mathbf{z}_t \in \mathbb{R}^{f \times h \times w \times c}$ is nosiy input, $\mathbf{z}_m \in \mathbb{R}^{f \times h \times w \times c}$ is masked video latent. Typically, $\mathbf{z}_t$, $\mathbf{z}_m$, and $\bar{m}$ are concatenated along the channel dimension as the denoising network's main stream inputs $\mathbf{z}_{in} \in \mathbb{R}^{f \times h \times w \times (c_m + 2c)}$. Both Minimax-Remover Zi et al. (2025) and EraserDiT Liu & Hui (2025) adopt the paradigm of channel-wise concatenation. Minimax-Remover is built on the pretrained text-to-video generation model Wan2.1-1.3B Wan et al. (2025). To accommodate its design, the original input channel size of the backbone model is expanded from 16 to 48 ($c_m = 16$), which inevitably requires full-parameter fine-tuning.

## 3 METHODOLOGY

### 3.1 PRELIMINARY

**Flow matching.** Flow matching video generative models employ a neural network to synthesize realistic video frames by learning a time-dependent vector field that guides samples from a noise distribution toward the target video distribution. Given an input video $\mathbf{x} \in \mathbb{R}^{f \times h \times w \times c}$, a pretrained variational autoencoder (VAE) encoder $\mathcal{E}$ encodes $\mathbf{x}$ to latent representation $\mathbf{z_0} = \mathcal{E}\left(\mathbf{x}\right)$. The rectified flow-matching loss

$$\mathcal{L}_\theta = \mathbb{E}_{t \sim p(t), \mathbf{z}, \mathbf{y}, c}\left[\|v_\theta\left(\mathbf{z}_t, t, \mathbf{y}, c_{text}\right) - \mathbf{v}_t\|_2^2\right], \tag{1}$$

where $\mathbf{z}_t$ is the linearly interpolated latent between $\mathbf{z_0}$ and noise $\epsilon \in \mathcal{N}\left(0, \mathbf{I}\right)$, $\mathbf{z}_t = (1-t)\mathbf{z_0} + t\epsilon$, $\mathbf{y}$ is the reference video latent, $c_{text}$ is text prompt, and the velocity $\mathbf{v}_t = \frac{d\mathbf{z}_t}{dt} = \epsilon - \mathbf{z_0}$.

### 3.2 MODEL ARCHITECTURE

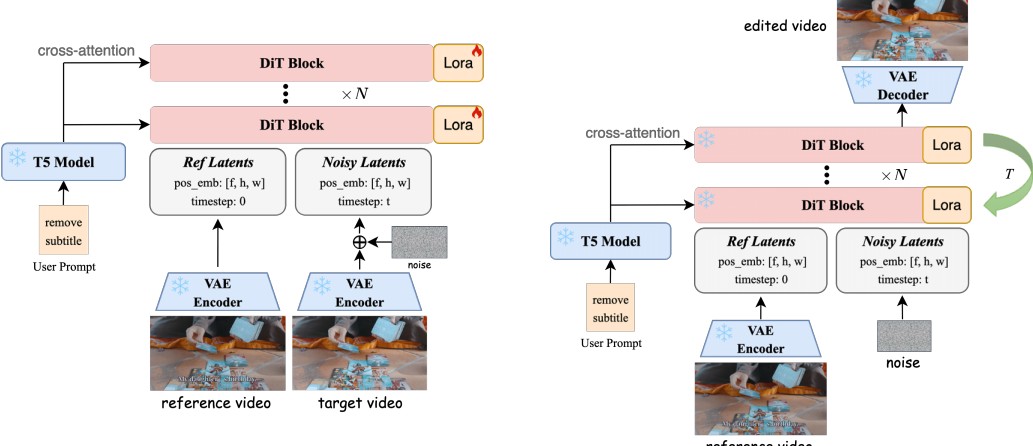

(a) The architecture of training phase.

(b) The architecture of inference phase.

Figure 2: The overview of our mask-free video subtitle erasure framework. We introduce a conditional video branch alongside the original video brach. Given the conditional video, the VAE encoder maps it into tokens, which are concatenated with video latent tokens and then sent to the DiT.

The overall framework is illustrated in Figure 2. We use LTX-Video-2B-0.9.6 HaCohen et al. (2024) as the video generation base model, which adopts a Diffusion Transformer (DiT) architecture. Our goal is to learn a model that can generate videos conditioned jointly on a text prompt and a reference videos. Given a reference video with subtitles, we first encode it into the latent space using the pre-trained VAE encoder. The reference video latents are patchified into $\mathbf{z}_{\text{ref}} \in \mathbb{R}^{B \times (F \times H \times W) \times C}$, same as the noisy video tokens. Since the reference video latents remain clean through the denoising process, we set their denoising timestep to 0 during diffusion. Then, the reference video tokens are

concatenated with noisy video tokens along the sequence dimension and processed jointly through successive DiT blocks. The input video tokens can be expressed as $\mathbf{z}_{in} = \mathbb{R}^{B \times 2(FHW) \times C}$.

We encode positional information via 3D Rotary Positional Embedding (RoPE) Su et al. (2024), where the embeddings for the reference $\mathbf{z}_{\text{ref}}$ are aligned with those of the noisy tokens $\mathbf{z}_{\text{noise}}$. Concretely, if a token position is denoted by the triplet $\mathbf{u} = (\text{f}, \text{h}, \text{w})$, then we set $\mathbf{u}_{\text{ref}} = \mathbf{u}_{\text{noise}} = (\text{f}, \text{h}, \text{w})$ for both reference video tokens and target video tokens. The primary motivation for this design is to ensure frame-by-frame alignment between the reference video and the target video.

Since subtitles occupy only a small portion of the overall video content and both the reference and target videos share identical positional encodings, directly applying Equation 1 to compute the loss tends to bias the model toward reconstructing the reference video. Therefore, we impose a higher loss penalty in the text regions to encourage the model to focus on subtitle removal. The focal loss function is defined by

$$\mathcal{L}_{focal} = \mathcal{L}_{\theta} * (\mathbf{I} + \alpha * \mathbf{M}_{subtitle}), \tag{2}$$

where $\mathbf{M}_{subtitle}$ indicates the mask obtained by filling the subtitle bounding box, $\mathbf{I}$ denotes an all-ones tensor with the same shape as $\mathbf{M}_{subtitle}$, and $\alpha$ is scalar.

### 3.3 DATA SYNTHESIS

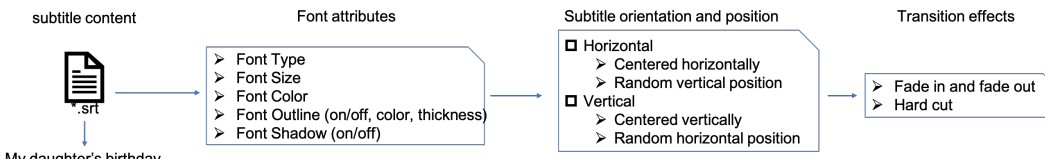

Figure 3: Data Synthesis pipeline.

To facilitate efficient model training, we simulate subtitles in the real-world videos through a rendering-based approach. To closely replicate the visual characteristics of subtitles in practical video scenarios, we construct a subtitle data synthesis pipeline, as illustrated in Figure 3. The entire pipeline comprises four components: (1) subtitle content acquisition; (2) font attribute configuration; (3) subtitle layout specification; and (4) transition effect definition for subtitle switching.

**Subtitle content acquisition.** We collected subtitles files (*.srt) from the internet spanning various genres, including films, anime, and television dramas. The dataset covers multiple languages, such as English, Chinese, Japanese, and Korean. We utilize the Python library *pysrt* to extract and parse the detailed content of the subtitle files.

**font attribute configuration.** We collected a range of commonly used Chinese and English font files, including *Source Han Sans*, *Source Han Serif*, *Alibaba PuHuiTi*, *Arial*, and *Helvetica*. We define the font size range as $[40pt, 150pt]$, and uniformly sample within this interval to obtain the current font size. Text colors are categorized into two distinct groups: bright-toned and dark-toned. Bright-toned text is defined by RGB values in which each channel—red, green, and blue—is independently sampled from the range $[231, 255]$, resulting in high-luminance color compositions. In contrast, dark-toned text comprises RGB values uniformly sampled from the range $[0, 230]$ across all three channels, yielding lower-luminance appearances. For bright-toned text, the font outline color is set to a dark hue, with RGB values sampled from the range $[0, 15]$. Conversely, for dark-toned text, the outline color is assigned a light hue, with RGB values drawn from the interval $[231, 255]$. The thickness of the text border is determined probabilistically: with a probability of $50\%$, it is set to zero, indicating no visible outline. For the remaining $50\%$, the border width is uniformly sampled from the interval $[2pt, 5pt]$, producing a visible outline with variable thickness. In certain cases, subtitle text is rendered with shadow effects. To simulate this visual feature, we apply a slight positional offset to the font and combine it with a light black color, thereby producing a shadow-like appearance beneath the primary text.

**Subtitle layout specification.** Subtitles are typically arranged horizontally in the lower region of the video frame, regardless of whether the video is in landscape or portrait orientation. However, vertically oriented subtitles do occur occasionally. Accordingly, we assign a probability of $75\%$

to horizontal layout and $25\%$ to vertical layout. When subtitles are arranged horizontally, they are centered along the horizontal axis while their vertical position is randomly assigned. Conversely, when subtitles are arranged vertically, they are centered along the vertical axis with a randomly determined horizontal placement.

**Transition effect definition for subtitle switching.** Subtitle transitions in video content typically occur via two primary mechanisms: (1) abrupt switching and (2) fade-in/fade-out effects. To simulate the latter, we modulate the text opacity over time, thereby achieving a smooth transitional appearance. Additionally, in music video (MV) scenarios, lyric text often exhibits dynamic color changes synchronized with temporal progression. We replicate this behavior to accommodate lyric erasure effects commonly observed in MV-style presentations.

### 3.4 LONG VIDEO INFERENCE

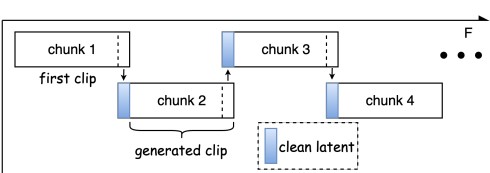
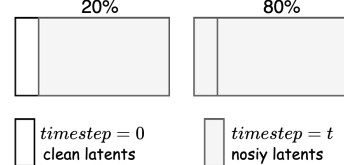

Figure 4: Long vidoe inference strategy.    Figure 5: First-frame conditioning.

In practical subtitle removal tasks, it is common to process videos exceeding 10 minutes in duration, such as television dramas and animated content. We begin by applying scene detection algorithms (e.g., PySceneDetect[1], TransNetV2 Soucek & Lokoc (2024)) to segment the entire video into individual shots, followed by shot-wise processing. For short shots ($\leq 5$ seconds), the model can handle them directly. However, for longer shots ($> 5$ seconds), we adopt a chunk-wise processing strategy. The chunk size is determined based on the video resolution: specifically, 121 frames for 720p videos, 65 frames for 1080p videos, and 41 frames for 1440p videos. To support the any-length video processing, we set the first-frame conditioning, as shown in Figure 5. During the training phase, the noised latent representation of the first frame is replaced with its clean counterpart with a probability of 0.2 and set the corresponding diffusion timestep to zero. During inference, except for the first chunk, the first frame of each subsequent chunk is initialized using the last frame of the preceding chunk. Experimental results demonstrate that overlapping by a single frame is sufficient to maintain satisfactory temporal consistency. To reduce memory consumption, frame data is written into a TS (Transport Stream) video file in a streaming manner upon completion of each shot.

To reduce the burden of complex prompt formulation during inference, we employ the fixed predefined prompt during training that omits the specific content of the subtitles. The prompt is defined as: *"Please remove the subtitle text from the video while preserving the character appearance, background composition, and color style. Do not add any new elements."* Since the current task is well-defined and the prompt remains unchanged, the prompt merely serves as a necessary input for the base model and has almost no impact on the video subtitle erasure task.

## 4 EXPERIMENTS

### 4.1 DATASETS

We collected $400,000$ high-definition, subtitle-free videos from the Pexels website[2] to serve as ground truth (GT) data. Embedded subtitle videos were subsequently generated on-the-fly using the data synthesis pipeline described in Section 3.3. Due to the absence of publicly available benchmark datasets for video subtitle removal, we construct a dataset of 400 samples to facilitate fair comparison across models. Each sample includes a clean video without subtitles, a corresponding

---

[1]`https://github.com/Breakthrough/PySceneDetect`
[2]`https://www.pexels.com/`

subtitle mask, and a version with embedded subtitles. In the dataset, videos with 720p resolution contain 121 frames, while those with 1080p resolution consist of 81 frames. We name this dataset the Video Subtitle Removal Benchmark (VSR-Bench-400) dataset.

## 4.2 IMPLEMENTATION DETAILS

We adopt Low-Rank Adaptation (LoRA) with rank 256. For the 2B-parameter LTX-Video-0.9.6 model, this adds just 381M trainable parameters ($19\%$ of the base model). Training uses a batch size of 32. For input resolutions close to $1080 \times 1920$, the input frame length is set to 65; for resolutions near $1280 \times 720$, the input length is set to 121. The training process comprises $100K$ iterations on 8 NVIDIA A800 (80GB) GPUs, utilizing AdamW optimizer with an initial learning rate of $2 \times 10^{-4}$. $\alpha$ is experimentally set to 5 in Equation 2. During inference, we experimentally use one sampling step without model distillation. This is primarily attributed to the strong conditional video input provided to the model, whereby regions outside the subtitle area are largely reconstructed through a *copy-paste mechanism*.

## 4.3 QUANTITATIVE EXPERIMENTS

We compare our method against Minimax-Remover Zi et al. (2025), a leading open-source video object removal approach based on the DiT architecture. Besides, we compare the representative GAN-based method Propainter Zhou et al. (2023) and the diffusion-UNet-based method Diffueraser Li et al. (2025). For visual quality assessment, we employ SSIM (Structural Similarity Index Measure), LPIPS(Learned Perceptual Image Patch Similarity) to measure frame-level fidelity. To assess the perceptual quality of subtitle removal results, we conducted a Mean Opinion Score (MOS) evaluation involving 20 human participants. As presented in Table 1, our proposed method consistently outperforms Minimax-Remover across all considered evalution metrics. Notably, our method exhibits significantly superior performance in both inference time and average MOS. We experimented with sampling steps of 1, 2, and 4. Based on objective metrics, both $step = 2$ and $step = 1$ have their respective advantages. Considering efficiency, we ultimately adopted the $step = 1$ configuration.

Table 1: Qualitative results on the VSR-Bench-400 dataset. "Time" indicates the average inference time per video with a resolution of $1920 \times 1080$ with 65 frames **without any acceleration method** on A800. "N/A" denotes not available. Mask-based approaches adopt **GT maskes**. The evaluation is conducted directly on the raw outputs of the model, without any post-processing operations added.

| Method | VSR-Bench-400 | | | | | |
| --- | --- | --- | --- | --- | --- | --- |
| | PSNR ↑ | SSIM ↑ | LPIPS ↓ | VFID ↓ | MOS ↑ | Time ↓ |
| Propainter | 26.8198 | 0.8230 | 0.1778 | 374.40 | N/A | 38s |
| DiffuEraser | 27.5084 | 0.8145 | 0.1222 | 303.05 | N/A | 166s |
| Minimax-Remover (6-step) | 28.3109 | 0.8785 | 0.1011 | 252.70 | 2.5 | 150s |
| Ours (SEDiT 4-step) | 31.2961 | 0.8783 | 0.1001 | 115.31 | N/A | 8s |
| Ours (SEDiT 2-step) | **31.5968** | **0.8805** | 0.0982 | 106.64 | N/A | 4s |
| Ours (SEDiT 1-step) | 31.5863 | 0.8805 | **0.0981** | **105.18** | **4.5** | **2s** |

## 4.4 QUALITATIVE EXPERIMENTS

As shown in Figure 6, the proposed method demonstrates effective and seamless subtitle removal. In the third column of the top-left example, despite the presence of pronounced subtitle-specific visual effects, our approach successfully eliminates the text without leaving visible traces. Such stylized and blurred subtitles are typically challenging for conventional OCR algorithms, which often suffer from low recognition accuracy and missed detections.

The bottom-left example highlights the robustness of our model: even though Russian subtitles were not included in the training data, the method still achieves high-quality removal. The top-right case further illustrates the generalization capability of our approach when handling colorful, bordered subtitles that occupy a large portion of the frame and obscure facial regions.

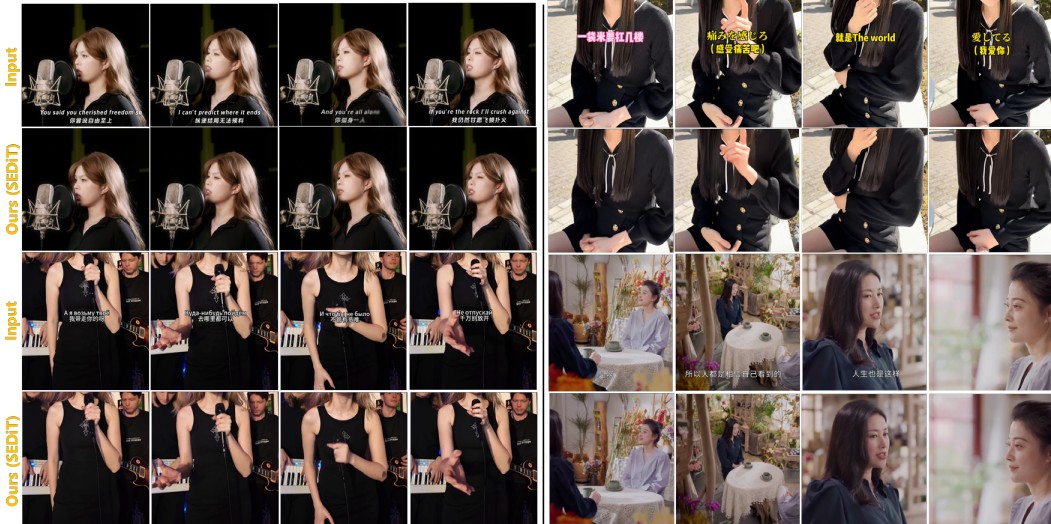

Figure 6: The evaluation results on multilingual subtitles include cases with mixed Chinese-English subtitles, mixed Chinese-Russian subtitles, Japanese subtitles, and Chinese-only subtitles. *Best viewed when zoomed-in.*

In the bottom-right example, the model excels at restoring complex textures such as patterned table-cloths. For frames without subtitles, our method accurately identifies the absence of text and refrains from performing unnecessary alterations. Overall, the results yield visually coherent reconstructions with minimal artifacts.

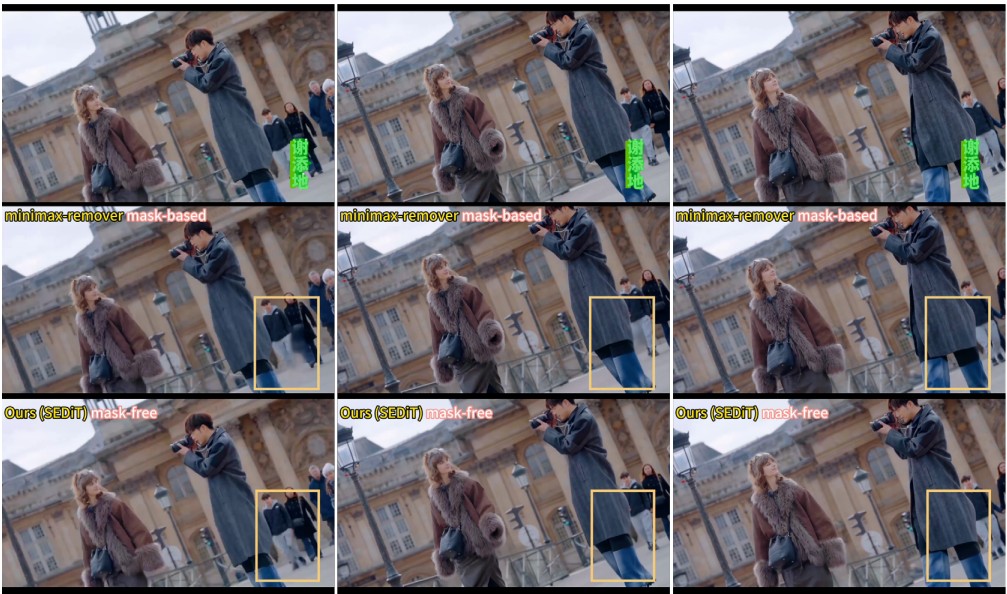

Figure 7: Visual comparison with state-of-the-art (SOTA) methods. The green highlighted region represents the subtitle mask, which is obtained using the SAM2 model. Focus on the visual comparison within the yellow boxes. *Best viewed when zoomed-in.*

To compare our approach with state-of-the-art mask-based video inpainting methods under real-world conditions, we utilize an interactive interface to obtain subtitle locations and apply SAM2 Ravi et al. (2024) for subtitle tracking. The resulting masks are then fed into the Minimax-Remover Zi et al. (2025) method to generate subtitle removal results. Compared to traditional OCR-based sub-

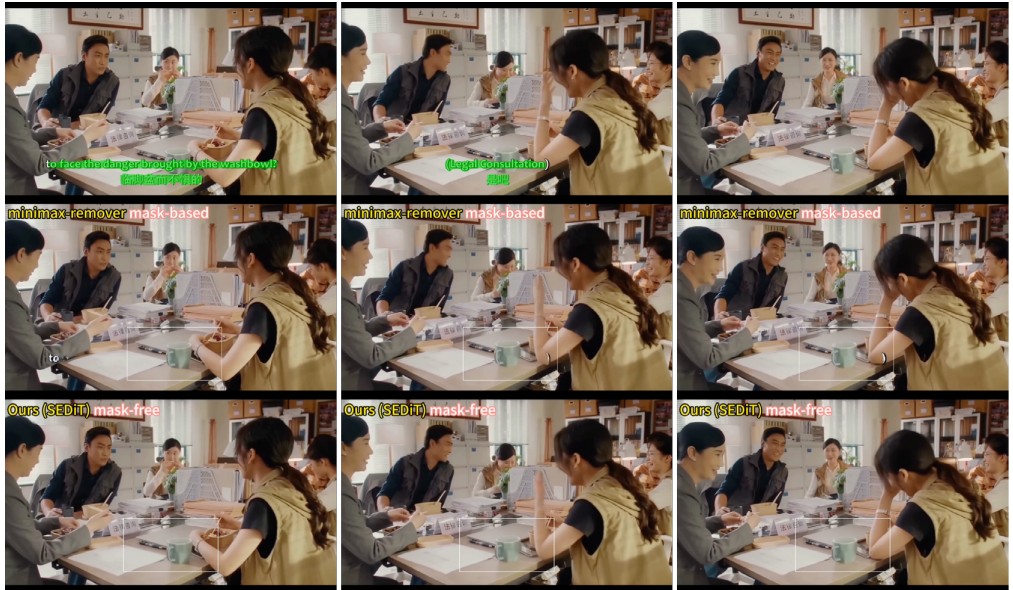

Figure 8: Visual comparison with state-of-the-art (SOTA) methods. The green highlighted region represents the subtitle mask, which is obtained using the SAM2 model. Focus on the visual comparison within the white boxes. *Best viewed when zoomed-in.*

title box extraction, this segmentation-based strategy yields more precise subtitle regions, which benefits mask-guided inpainting methods.

As illustrated in Figure 7, our mask-free approach successfully reconstructs occluded body parts of distant characters, whereas the baseline method introduces noticeable blurring artifacts. The final column of Figure 7 further shows that, in cases of mild occlusion, the baseline method can still produce reasonably good reconstructions.

As shown in Figure 8, when the subtitle mask is inaccurately defined, mask-based methods are prone to error propagation. Comparing the second and third columns, the incomplete mask in the second column results in residual subtitle artifacts in the generated output. This error further propagates to the third column, even though the input frame in that case contains no subtitles. **Additional high-resolution visual results are provided in the supplementary material.**

## 4.5 ABLATION STUDY

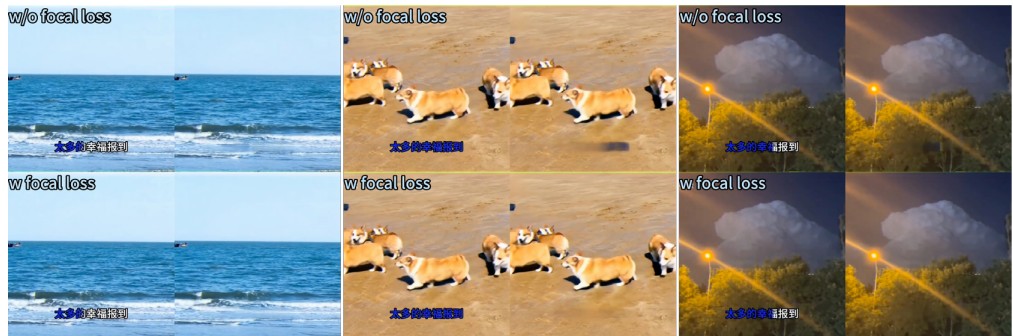

Figure 9: Ablation study for the focal loss. "w/o focal loss" denotes "without focal loss". "w focal loss" indicates "with focal loss".

**Effectiveness of the focal loss.** We perform an experiment by setting $\alpha = 0$ in Equation 2, which indicates directly using the original conditional rectified flow-matching loss to train our model. The

experiment presented in Figure 9 indicates that this approach fails to completely remove subtitles when dealing with complex styles such as gradient-colored text. By incorporating focal loss, the model is encouraged to concentrate more effectively on subtitle regions, resulting in significantly improved removal performance.

# 5 LIMITATIONS

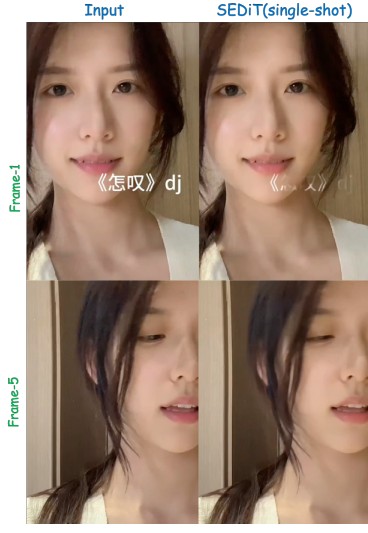

(a) Visual result of SEDiT processing the video clip spanning frames [1, 4] in isolation.

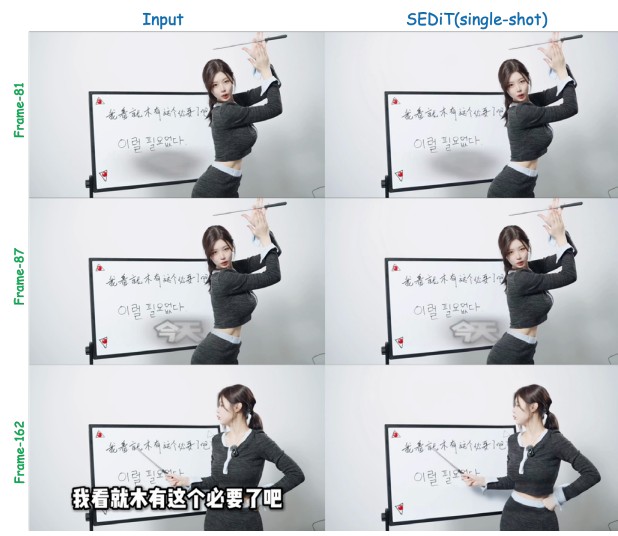

(b) Visual results of processing severely motion-blurred text.

Figure 10: The failure case of the SEDiT.

While SEDiT demonstrates strong performance in removing subtitles across most video scenarios, there are still some limitations. Specifically, we found that SEDiT may fail to fully remove static subtitles in extremely short shots (less than 10 frames). As shown in Figure 10a, the first shot contains only 4 frames. The single-shot subtitle removal (first row, second column) fails to completely eliminate the subtitles. In addition, we observe that the current model is unable to remove text with severe motion blur, as ilustrated in Figure 10b(the first two rows). Nevertheless, for typical clear subtitles, satisfactory removal results can be obtained even with relatively large font sizes.

# 6 CONCLUSION

We propose **SEDiT**, a lightweight, one-stage, mask-free video editing framework for high-field video subtitle erasure. Unlike previous mask-based video inpainting methods, our approach eliminates the need for explicit masks, thereby bypassing the video mask segmentation process entirely. Furthermore, our method preserves the architecture of the underlying generative model, enabling efficient fine-tuning via LoRA. This design choice also allows for seamless integration with higher-quality video generation backbones to further enhance performance.

To support subtitle removal in videos of arbitrary length, we adopt a chunk-wise processing strategy, dynamically adjusting the chunk size based on the input resolution. To mitigate temporal discontinuities across chunks, we probabilistically inject the initial frame as a conditioning signal. Benefiting from this strong conditional guidance, frame repetition across chunks achieves temporal consistency with minimal artifacts.

During inference, our method requires only one step to produce high-quality results. Leveraging the high compression ratio of the base VAE model, our approach can directly process 1080p resolution videos, completing 65 frames in just 2 seconds—making it particularly well-suited for large-scale deployment.

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

## A APPENDIX

### A.1 ETHICS STATEMENT

The research conducted in the paper conforms, in every respect, with the ICLR Code of Ethics.

### A.2 REPRODUCIBILITY STATEMENT

We have provided implementation details in Section 3.3 and Section 4.

### A.3 THE USE OF LARGE LANGUAGE MODELS (LLMS)

Large Language Models (LLMs) were used solely for polishing writing. They did not contribute to the research content or scientific findings of this work.

### A.4 THE VISUAL COMPARISONS OF DIFFERENT INFERENCE STEPS

Video subtitle removal is a strongly constrained video editing task in which the input and target videos differ only slightly. Our experiments show that satisfactory results can be achieved with a limited number of sampling steps during inference. Therefore, we investigate the effect of sampling step count on the performance of subtitle removal. As illustrated in Figure 11, setting the sampling steps to 1, 2, or 4 has minimal impact on the visual results. The configuration with $step = 1$ effectively restores complex clothing textures.

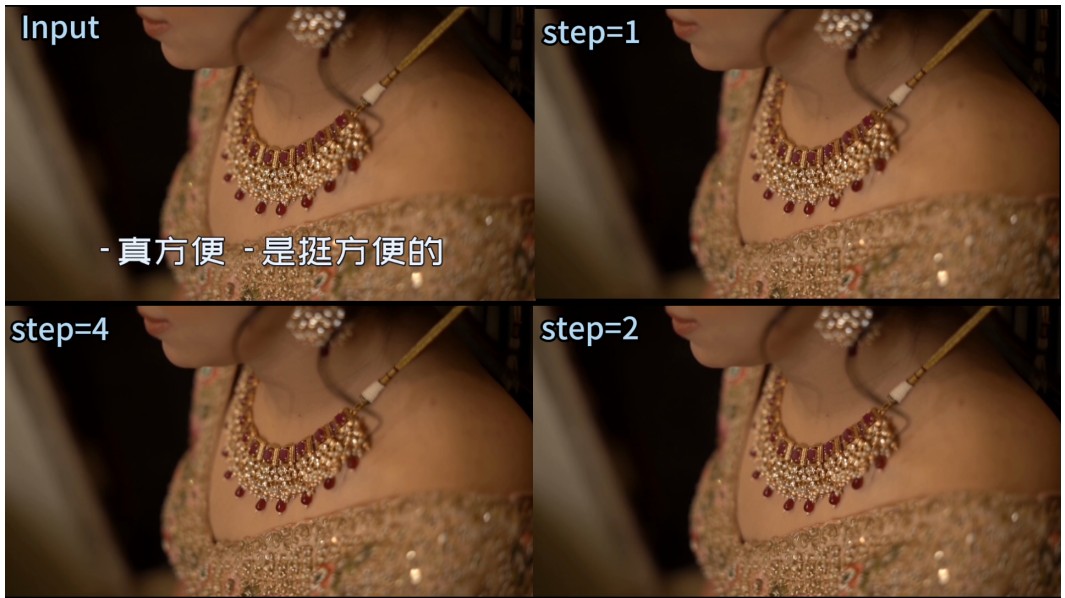

Figure 11: Comparison of visual effects under different sampling steps.

## A.5 THE VISUAL RESULTS FOR VIDEO STYLE TRANSFER

In the Figure 12, "Remove watermark" corresponds to the user prompt: "Please remove the **watermark** from the video while preserving the character appearance, background composition, and color style. Do not add any new elements." "Remove subtitle" indicates the use of a predefined prompt, while "None" means the prompt is an empty string ("")." The results suggest that the subtitle removal prompt has little impact on the output. This is because the training process was designed as a single-task setup.

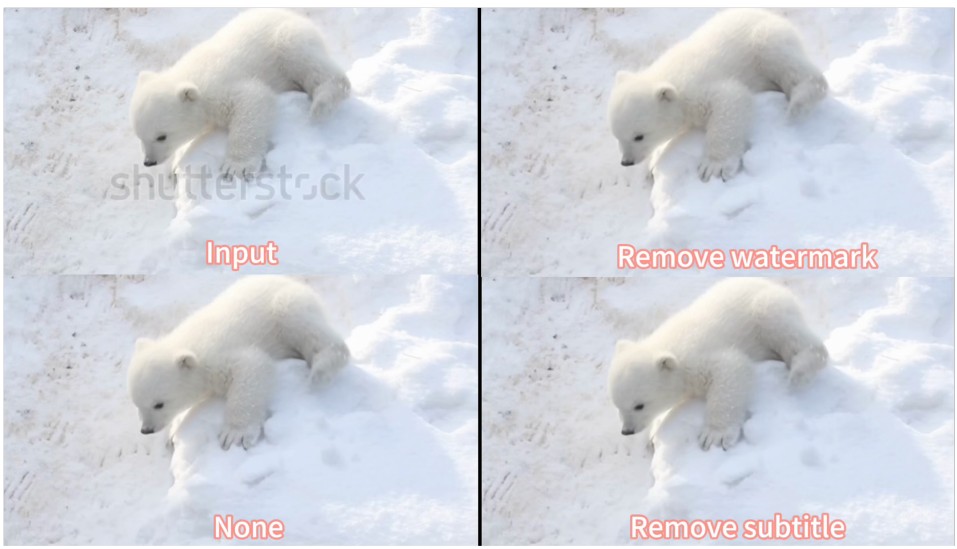

Figure 12: Results of watermark removal.

To evaluate the generalization capability of the SEDiT architecture and the effectiveness of user prompts, we train a video stylization model on the recently released video editing dataset Ditto-1M Bai et al. (2025a). As shown in Figure 13, the **user prompt** precisely guides the model to generate distinct visual styles. This demonstrates that our model architecture can be readily adapted to other video editing tasks.

## A.6 FAILURE CASE OF OCR

As shown in Figure 14, we apply the current state-of-the-art OCR algorithm, PaddleOCR-VL, to detect text in videos containing background characters. In addition to subtitle content, the algorithm also detects other textual elements present in the scene. As shown in the result of the second row, first column, PaddleOCR-VL fails to effectively recognize subtitles with transition effects. Such detection errors directly lead to the failure of mask-based subtitle removal methods. In contrast, our approach successfully removes these subtitles while preserving the original textual content in the scene.

## A.7 MORE VISUAL COMPARISONS ON VSR-BENCH-400 DATASET

We further compared the visual performance of SEDiT with Propainter Zhou et al. (2023), DiffuEraser Li et al. (2025), and Minimax-Remover Zi et al. (2025) on the VSR-Bench-400 dataset. As shown in Figures 15, 16, 17, Propainter tends to produce blurriness or artifacts when subtitles occupy a large proportion of the frame. DiffuEraser alleviates some artifacts, but still generates structurally inconsistent content. Among the mask-based approaches, Minimax-Remover achieves the best overall performance, though blurry regions remain. Overall, our method strives to preserve as much of the original video content as possible, yielding visually satisfactory results.

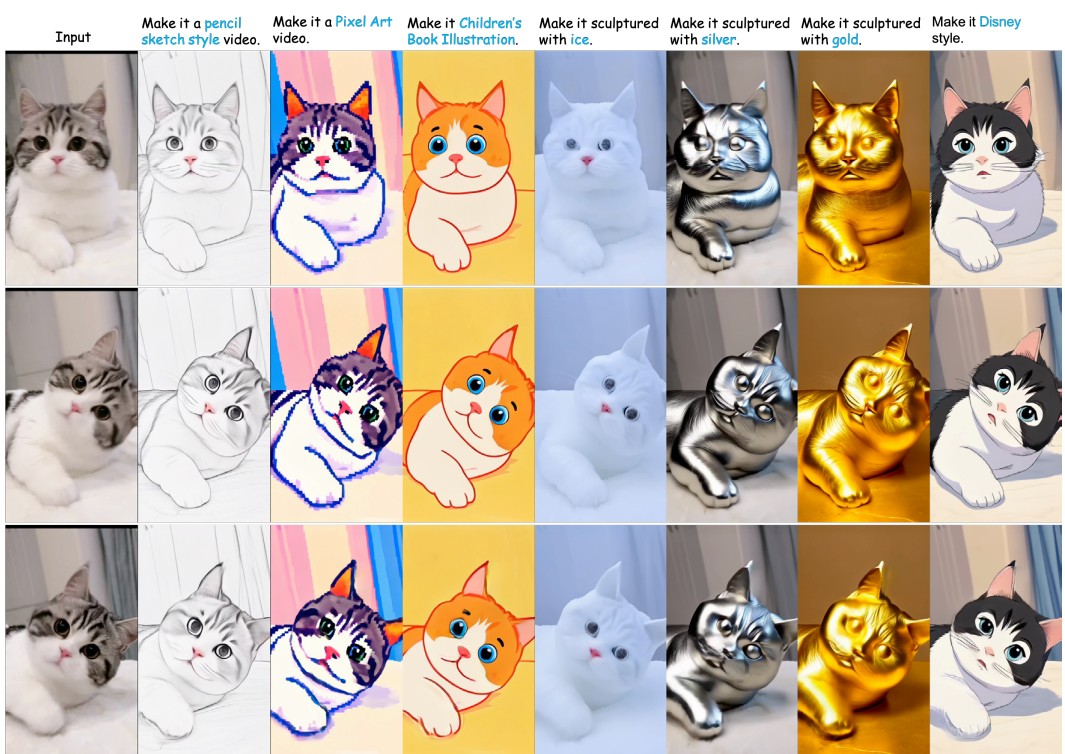

Figure 13: Visual Results for video style transfer.

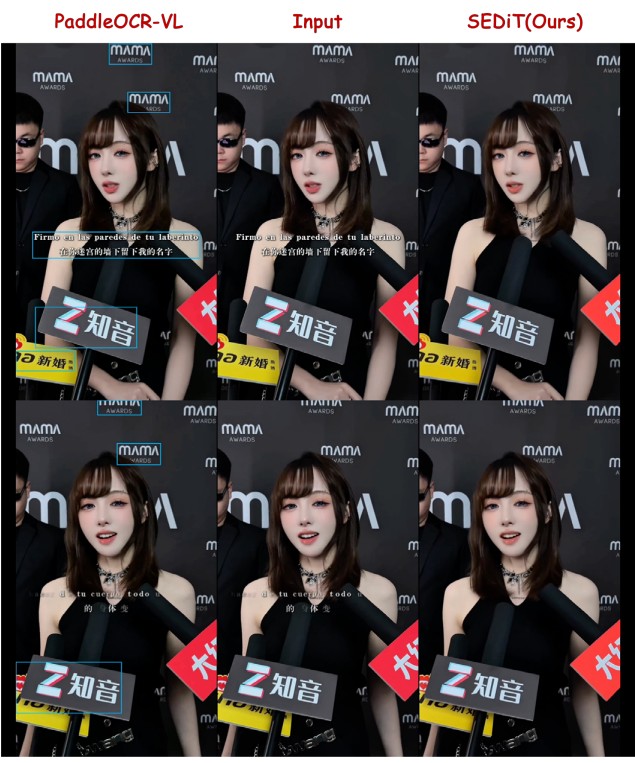

Figure 14: The OCR failure case in subtitle recognition. *Best viewed when zoomed-in.*

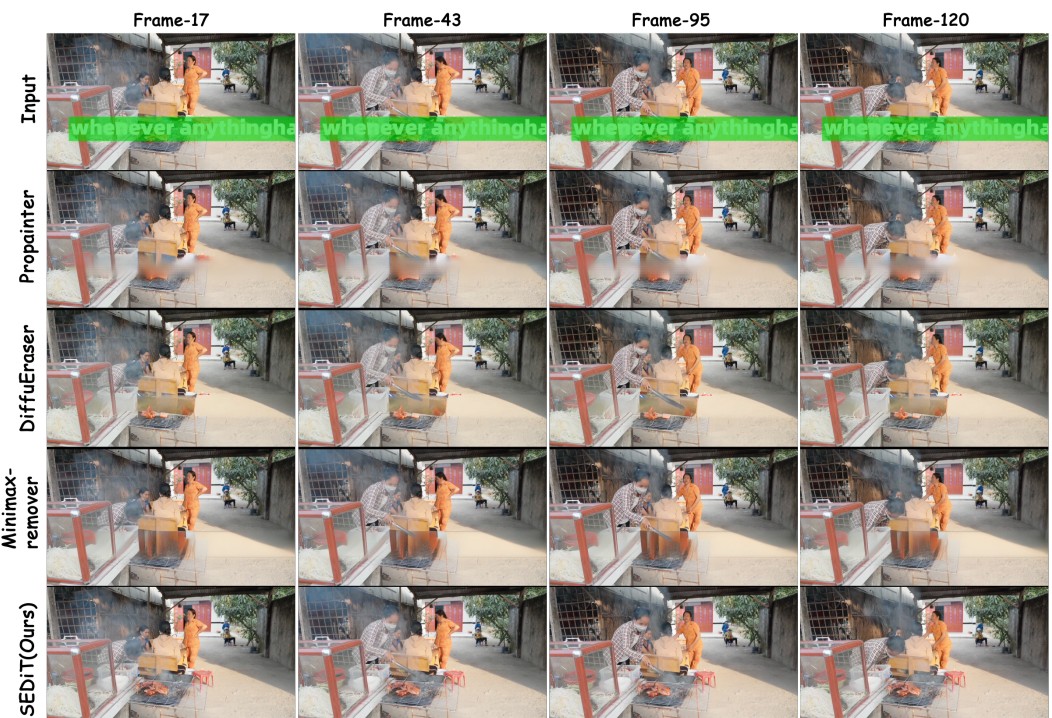

Figure 15: The visual comparison on VSR-Bench-400 dataset. The green highlighted region represents the subtitle mask, which is Ground-truth (GT) masks generated by filling the subtitle boundaries.

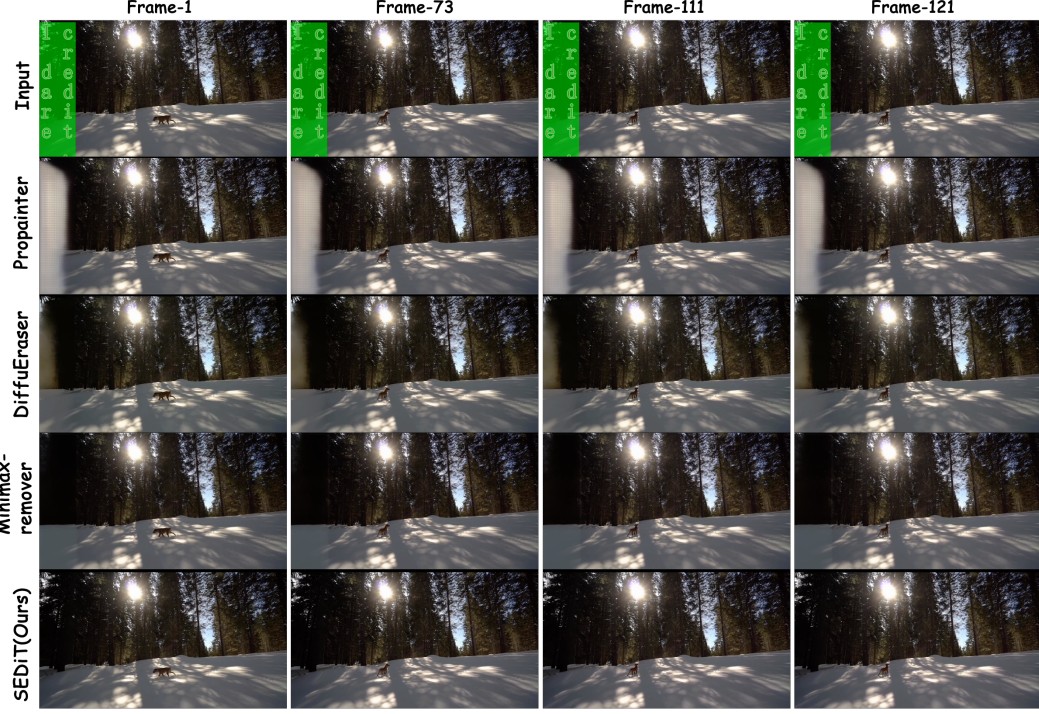

Figure 16: The visual comparison on VSR-Bench-400 dataset. The green highlighted region represents the subtitle mask, which is Ground-truth (GT) masks generated by filling the subtitle boundaries.

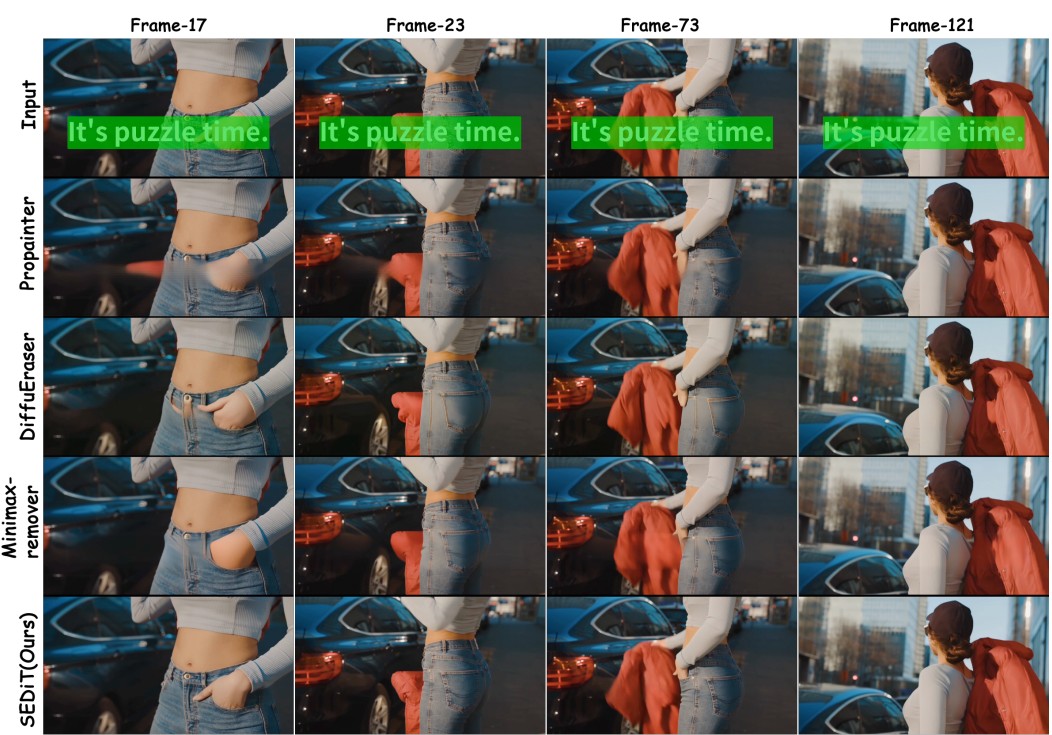

Figure 17: The visual comparison on VSR-Bench-400 dataset. The green highlighted region represents the subtitle mask, which is Ground-truth (GT) masks generated by filling the subtitle boundaries.

