# OpenReview forum: "SEDiT: Mask-Free Video Subtitle Erasure with Prompt Instruction"
_ICLR.cc/2026/Conference — Submitted to ICLR 2026_

### Official Review · Reviewer_Hrxk · 2025-10-31

**Soundness:** 3
**Presentation:** 2
**Contribution:** 3
**Rating:** 2
**Confidence:** 4

**Summary:**

This paper introduces SEDiT, a DiT–based framework for inference-time mask-free video subtitle removal. Instead of relying on optical character recognition (OCR) and the inpainting mask, SEDiT performs subtitle erasure through prompt-guided conditioning within a pretrained video generative backbone (LTX-Video). The model processes latent representations of subtitle-containing videos and reference videos by concatenating them along the sequence dimension, allowing training and inference without architectural modification to the original video DiT backbone. A rendering-based data synthesis pipeline is proposed to create large-scale paired data, covering diverse font attributes, layouts, and transition effects for realistic subtitle simulation. To handle long videos, the authors use a chunk-wise inference strategy with single-frame conditioning to maintain temporal coherence.

The claimed contributions are:
- A mask-free, prompt-guided framework for video subtitle erasure using a pretrained DiT backbone;
- A sequence-concatenation conditioning strategy enabling fine-tuning without changing model structure;
- A chunk-wise processing mechanism for arbitrary input video length and a single-first-frame condition for mitigating temporal discontinuity.

**Strengths:**

- The method should be easy to generalize to different DiT backbones since it does not require architectural changes, which makes the method adaptive to better DiT backbones in the future to achieve better results.
- The paper shows a carefully engineered long-video processing pipeline — integrating shot detection, adaptive chunking, and first-frame conditioning to maintain temporal coherence across segments. While such components have appeared individually in prior works (e.g., ProPainter, VideoCrafter, Stable Video Diffusion), the authors present a more systematic and reproducible implementation suitable for large-scale or industrial scenarios.
- Tab.1 demonstrates noticeable inference-time efficiency, which is valuable for long-video processing.
- The data synthesis pipeline is comprehensive, considering diverse font attributes, layouts, and transition effects for realistic subtitle simulation.
- A key practical strength is the mask-free inference pipeline, which avoids the need for external OCR or segmentation models. This design reduces error propagation and simplifies large-scale deployment, although supervision masks are still used during training.

**Weaknesses:**

- I am worried that the second and third points of contributions might be overclaimed. The second (“sequence-wise concatenation of conditional video without architectural modification”) directly follows the conditioning design of FLUX-Kontext and Qwen-Image, representing an adaptation rather than a new method. The third (“chunk-wise processing with first-frame conditioning”) reflects practical engineering design for long videos, not algorithmic innovation. The paper should clarify this hierarchy to avoid overclaiming.
- The paper does not clarify whether the evaluation videos in VSR-Bench were synthesized using the same rendering pipeline, subtitle styles, or font templates as the training corpus of 400k synthetic videos. If the training and evaluation sets share generation parameters, the model could implicitly overfit to the rendering artifacts rather than generalizing to unseen subtitle styles. Moreover, the authors do not state whether the compared baseline (Minimax-Remover) was retrained or fine-tuned on the same synthetic dataset. Without ensuring consistent data exposure, the comparison risks favoring the proposed model, making it difficult to assess true generalization performance or fairness across methods.
- Only Minimax-Remover is compared. Recent diffusion-based open-source baselines such as [1][2] or strong inpainting backbones like ProPainter are omitted. Including these would make the comparison credible.

[1] Towards Language-Driven Video Inpainting via Multimodal Large Language Models (CVPR'24)
[2] Any-length Video Inpainting and Editing with Plug-and-Play Context Control (SIGGRAPH'25)

**Questions:**

- The paper does not explicitly claim that the mask-free variant outperforms mask-based approaches, but it repeatedly suggests qualitative robustness advantages (e.g., in motivation and qualitative results sections). The argument that mask-based methods are prone to error propagation under imperfect masks is reasonable, yet no controlled experiment isolates whether the observed improvements come from the mask-free formulation itself or the more advanced DiT backbone. A stronger evaluation would include a mask-based baseline built on the same DiT backbone and dataset to clarify whether the benefits arise from the absence of masks or from other architectural or training factors.
- The fixed-format prompt (“remove subtitles in the video”) is manually chosen, and its functional role is unclear. Does the prompt still function as semantic guidance? Meaning that changing the prompt to instruct other tasks (like "remove watermark in the video") would result in a different application. Or is it just a condition token now? Then I wonder how much it contributes to the final performance, and is it really necessary since the paired data should make the model understand the task enough? The paper would be more comprehensive if the authors could include comparisons among (i) fixed-prompt, (ii) prompt-free, and (iii) learnable-token (as in Minimax-Remover) settings to clarify prompt contribution.
- Some terms are potentially overstated. For example, the method is only "mask-free" during inference time, and the pipeline is not technically "end-to-end" since it requires external shot-boundary detection and chunk-wise processing with first-frame reinjection. The authors may consider rephrasing to avoid misunderstanding.
- The commonly used metric, VFID, to assess video quality/perception in the literature is missing.
- The paper asserts that stylized or blurred subtitles defeat OCR detectors but provides no detection statistics or visual comparisons. Reporting detection recall/F1 of representative OCR algorithms on the same data would substantiate this statement.

Minor writing issues:
- The paper constantly uses \citet for all citations. Use \citet{} when the citation is part of the sentence, e.g., “Zhou et al. (2023) propose xxxxx.” and \citep{} when the citation is in parentheses, e.g., “Video inpainting has been widely studied (Zhou et al., 2023).”.
- L72: OCR abbreviation first appears, but no full name is given.
- Fig.6, left bottom case third column, I think the input and ours show different frames.
- In Fig.7 and 8, what does the bounding box mean? And what do the green highlights mean? The captions should explain the notations on the figures more clearly.
- In the supplementary material, I think the third clip of demo 1 mixed the input and output videos.
- typos: precessing (L92), The reference video latents are patchified ... as the noisy video tokens (L205), Firt-frame conditionin (Fig.5 caption)

---

> ### Author Response · Authors · 2025-11-20
>
> Response to the Questions:
> 1. In Figure 8, the mask in the second column is imperfect (as highlighted by the white box), which leads to artifacts in the subtitle removal results produced by Minimax-remover. The third column shows input videos without subtitles, yet Minimax-remover introduces error propagation in its output. Although Minimax-remover uses the Wan2.1-1.3B model (VAE compression rate: $4 \times 8 \times 8$), its performance is comparable to LTX-Video-0.9.6 (VAE compression rate: $8 \times 32 \times 32$). Wan2.1-1.3B often yields better generation quality at the cost of slower speed.
>
> The Appendix includes an case (Figure 14) where OCR detection is inaccurate. In such scenarios, mask-based methods may mistakenly erase non-subtitle text in the background, whereas our approach effectively preserves these non-subtitle elements. This highlights the robustness of mask-free methods. One key advantage of our one-stage model is its automatically distinguish between subtitle and non-subtitle text based on temporal variations.
>
> Moreover, mask-based models require modifying the input channels of the base video generation architecture, which necessitates full-parameter fine-tuning and significantly longer training time to achieve convergence.
>
> 2. Fixed prompts are used because the objective of subtitle removal is clearly defined and does not require describing the video content, making the system easier for users to operate.
>
> Fixed prompts are adopted because the objective of subtitle removal is clearly defined and does not require describing the video content, making the system easy for users to operate. We conducted an additional set of experiments comparing: "remove  subtitle",  "remove watermark" and empty prompts. As shown in Figure 12, the difference in performance is negligible, indicating that prompts have minimal impact in this task. In this context, prompts are mainly used to avoid modifying the base model architecture, thereby reducing training costs.
>
> Moreover, we have extended our current method, SEDiT, to the domain of video stylization. As illustrated in Figure 13, different styles can be effectively controlled through prompts, highlighting the importance of prompt design in broader video editing applications.
>
> 3. We thank the reviewer for pointing this out, which has helped us improve the precision of our wording. We have revised "mask-free training" to "mask-free inference" for clarity. As shown in Figure 1, the term "end-to-end" was originally used to indicate that our method eliminates the need for a separate subtitle detection step. To handle long videos, shot transition detection and chunk-wise processing are essential components, regardless of the specific subtitle removal algorithm used. To avoid ambiguity, we have replaced "end-to-end" with "one-stage" in the revised manuscript.
>
> 4. We have added the VFID metric. In addition, we expanded the original test set from $200$ to $400$ videos and re-evaluated the objective metrics accordingly. For the comparison, Minimax-Remover was provided with ground-truth subtitle boxes as masks.
>
> 5. We analyzed a representative video using the current state-of-the-art OCR method, paddleOCR-VL[1]. As shown in Figure 2, the text boxes detected by the OCR model cannot distinguish between subtitle text and other on-screen text. This is because current OCR methods lack advanced semantic understanding capabilities. In contrast, our method successfully removes subtitles while preserving other textual elements in the video.
>
> Minor writing issues:
>
> 1. The errors have been corrected in the revised version, and we now use \citet{} and \citep{} appropriately.
>
> 2. After double  check, we confirmed that it is indeed the same frame. The current VAE with high temporal compression (factor of 8) struggles to reconstruct high dynamic content effectively. We have replaced it with a clearer case to better illustrate the model’s reconstruction capability. Overall, the process is lossy.
>
> 3. The green highlights indicate the subtitle regions to be removed, which are generated by the SAM2 model. The bounding boxes in Figures 7 and 8 are provided to guide readers in focusing on the visual differences within those regions.
>
> 4. In the supplementary material, the third clip of demo 1 mixed the input and output videos. We have re-uploaded the corrected video. Please check it.
>
> 5. All spelling errors have been corrected.
>
> [1] Cui, Cheng, et al. "PaddleOCR-VL: Boosting Multilingual Document Parsing via a 0.9 B Ultra-Compact Vision-Language Model." arXiv preprint arXiv:2510.14528 (2025).

---

> > ### Author Response · Authors · 2025-11-21
> > **Response to the Weaknesses:**
> >
> > 1. Regarding the second contribution: our architecture is adapted from image editing methods, as detailed in the Introduction. Some recently proposed video editing approaches, such as Ditto [3] and Lucy Edit [4], adopt different strategies. Ditto is built upon the in-context video generator from VACE [5], which functions as an additional plugin similar to ControlNet. Lucy Edit, on the other hand, introduces the reference video through channel-wise concatenation. Both approaches differ from the method we adopt.
> >
> > 2. Regarding the third contribution: we have clarified in the revised version that this is a simple and effective strategy under strong reference conditioning.
> >
> > 3. We reconstructed the evaluation dataset to ensure that the video sources in VSR-Bench-400 are different from those used in the training data. Regarding the subtitle rendering pipeline, VSR-Bench-400 does adopt the same rendering process as the training set. However, our data synthesis pipeline includes numerous randomized operations—such as random subtitle selection and random font attribute assignment—to ensure diversity in the synthetic data. Additionally, Figures 6, 7, 8, 9, 10, 12, and 14 showcase real-world videos with naturally embedded subtitles, while the results in Figures 11 and 15 are from the VSR-Bench-400 dataset.
> >
> > 4. We evaluated the performance of Minimax-Remover on the VSR-Bench_400 dataset using ground-truth masks (GT masks), ensuring that differences in subtitle styles do not affect the performance of this mask-based removal algorithm.
> >
> > 5. Since Minimax-Remover currently demonstrates outstanding performance in video removal tasks, we chose to compare only this mask-based method. ProPainter is a Transformer+GAN-based approach, and according to the papers of Minimax-Remover, DiffuEraser, and EraserDiT, its performance is inferior to diffusion-based methods. The video inpainting method in [1] adopts a U-Net diffusion model, while [2] is built upon CogVideoX and focuses more on object generation rather than removal.
> >
> >  [1] Wu, Jianzong, et al. "Towards language-driven video inpainting via multimodal large language models." Proceedings of the IEEE/CVF Conference on Computer Vision and Pattern Recognition. 2024.
> >
> > [2] Bian, Yuxuan, et al. "Videopainter: Any-length video inpainting and editing with plug-and-play context control." Proceedings of the Special Interest Group on Computer Graphics and Interactive Techniques Conference Conference Papers. 2025.
> >
> > [3] Bai, Qingyan, et al. "Scaling Instruction-Based Video Editing with a High-Quality Synthetic Dataset." arXiv preprint arXiv:2510.15742 (2025).
> >
> > [4] Lucy Edit: Open-Weight Text-Guided Video Editing
> >
> > [5] Jiang, Zeyinzi, et al. "Vace: All-in-one video creation and editing." arXiv preprint arXiv:2503.07598 (2025).

---

> > > ### Comment · Reviewer_Hrxk · 2025-11-21
> > >
> > > Dear authors,
> > >
> > > Thanks for addressing my doubts in the Weaknesses part. The clarification on contributions seems fine. However, my two main concerns are not solved.
> > >
> > > - The authors did not answer whether the Minimax-Remover baseline was retrained or finetuned on the proposed training dataset to avoid a potentially unfair comparison.
> > > Though there are randomized operations when generating the training and evaluation sets, the subtitle rendering pipelines are the same. Using the same rendering pipeline (even with randomizations) still creates a significant risk that your model is primarily learning to reverse or 'undo' the specific artifacts and styles introduced by your own synthesis pipeline. This makes the VSR-Bench benchmark results potentially biased towards your method.
> > > You mentioned evaluating Minimax-Remover using ground-truth masks (GT masks) on the VSR-Bench_400 dataset. However, I did not find any results in the paper stating this. Also, this is not a standard evaluation practice. A fairer comparison could be achieved by ensuring both models are exposed to the same data distribution, e.g., by retraining or finetuning Minimax-Remover on your synthetic data.
> > >
> > > - I find the decision to compare against only one baseline (Minimax-Remover) and to exclude others unjustified. The comparison must be strengthened by including other recent, high-performing video inpainting/editing methods. Your claim that ProPainter's performance is inferior, based on external comparisons, is not sufficient. The paper needs to directly compare against other SOTA methods in the broader video inpainting community (such as ProPainter or other strong diffusion-based approaches mentioned by me, Reviewer 5uKc, and Reviewer 2h8Z) to credibly validate the superiority of your proposed framework. Excluding these crucial comparisons makes the evaluation appear selective and less comprehensive.

---

> > > > ### Author Response · Authors · 2025-11-24
> > > > **Response to the Weakness parts**
> > > >
> > > > Dear Reviewer,
> > > >
> > > > Regarding the two concerns you mentioned:
> > > >
> > > > 1. Minimax-Remover was not retrained or fine-tuned on the proposed training dataset. We did not fine-tune the baseline model for two reasons:
> > > >
> > > > (1) Minimax-Remover has only released the inference code and model, without providing the training or fine-tuning code.
> > > >
> > > > (2) For the mask-based methods, we use subtitle boundary filling to generate rectangular masks (GT masks). The masked video input to the model ($V_{masked} = V \cdot (1 - \mathbf{M}) + \mathbf{0} \cdot \mathbf{M}$) does not contain visible subtitles and is independent of the subtitle rendering process. In addition, the reconstructed VSR-Bench-400 dataset employs original videos different from those used in training, specifically to avoid unfair comparisons cased by potential data leakage.
> > > >
> > > > Regarding the use of GT masks in mask-based methods, we have added the description to the caption of Table 1.
> > > >
> > > > 2. To provide  a more comprehensive demonstration of the superiority of the proposed framework, we have added comparative results of Propainter and DiffuEraser on the VSR-Bench-400 dataset. Although the optimal processing resolution for these two methods is not 1080p, we nevertheless enforced the models to process 1080p videos. All results are directly generated by the models without any post-processing. The comparative results are presented in the Table below.
> > > >
> > > > | Method     | PSNR ↑ | SSIM ↑ | LPIPS ↓ | VFID ↓ | Time ↓ |
> > > > |------------|--------|--------|--------|--------|--------|
> > > > | Propainter     | 26.8198  | 0.8230  | 0.1778 | 374.40 | 38s |
> > > > | DiffuEraser   | 27.5084  | 0.8145  | 0.1222 | 303.05 | 166s |
> > > > | Minimax-Remover (6-step) | 28.3109  | 0.8785  | 0.1011 | 252.70 | 150s |
> > > > | SEDiT (1-step) |  31.5863 | 0.8805  | 0.0981 | 105.18 | 2s |
> > > >
> > > > It can be observed that our method demonstrates a significant advantage in terms of the FVD metric. In addition, we have supplemented visual comparisons in the appendix, as shown in Figures 15, 16, and 17.

---

> > > > > ### Comment · Reviewer_Hrxk · 2025-11-25
> > > > >
> > > > > Sincerely appreciate the authors' efforts in addressing my main concerns. The explanation of GT masks usage is critical. Though it is not a common practice, it is indeed strong proof of SEDiT not gaining performance improvement from baselines' suboptimal mask predictions. More baselines are great for making the results more convincing.
> > > > >
> > > > > All in all, I believe the paper is in much better shape now. I am happy to raise my score.

---

> > ### Comment · Reviewer_Hrxk · 2025-11-21
> >
> > Dear authors,
> >
> > Thanks for the thorough answers to my questions and the effort in the additional experiments. The clarification on mask-free vs. mask-based methods is reasonable. The rephrasing of certain concepts makes the statements clearer and less misleading. An additional VFID metric makes the results stronger. And the OCR failure further comprehends the paper motivation.
> >
> > However, I still have two concerns:
> > - Since the results in Figure 12 show that prompt instructions have minimal impact on the task, you may want to revise the title ("xxx with Prompt Instruction") and framing to accurately reflect that the prompt is a fixed/dummy token for the DIT structure, and not a functional instruction interface for the core task. In the context of generative models, including the phrase "with Prompt Instruction" in the title typically implies that the system can be directed to perform different tasks or yield varied outputs by providing distinct textual prompts.
> > - The explanation of notations in Figures 7 and 8 should be added to the caption to make it more readable.

---

> > > ### Author Response · Authors · 2025-11-24
> > >
> > > We sincerely appreciate your constructive feedback and the time you have taken to review our manuscript. Your comments have been very helpful in improving the clarity and quality of our work. Below we address the specific concerns you raised:
> > >
> > > 1. In the current video subtitle erasure task, the prompt indeed only serves as a placeholder. We have revised the title to "SEDiT: Mask-Free Video Subtitle Erasure via Diffusion Transformer", and the corresponding statements in the manuscript have been updated accordingly.
> > >
> > > 2. The symbol explanations for Figures 7 and 8 have now been supplemented in the revised paper.

---

> > > > ### Comment · Reviewer_Hrxk · 2025-11-25
> > > >
> > > > Thank the authors for refining these small issues. The new title is great! No misleading terms, and the main concept of DiT is mentioned. Clearer captions of Figures 7 and 8 make it much more readable.
> > > >
> > > > But I still notice some misuse of citations in the revised paper. The authors may want to check how those citations in LaTeX should be used to improve the writing quality.

---

### Official Review · Reviewer_5uKc · 2025-11-01

**Soundness:** 2
**Presentation:** 3
**Contribution:** 2
**Rating:** 4
**Confidence:** 3

**Summary:**

This manuscript presents an end-to-end video subtitle erasure model based on a Diffusion Transformer, eliminating the need for additional masks for the subtitle region. More specifically, the authors concatenate the source video latents with noise along the sequence dimension and input them into the DiT backbone, avoiding modification of the model architecture and allowing it to be applied to other video backbone models. During training, the authors applied a focal loss to impose a higher loss penalty on the text regions. Meanwhile, the authors also use first-frame conditioning to enable chunk-wise long video processing. In addition, the authors propose a data synthesis pipeline to simulate various subtitles in real-world videos. From the experimental results, the proposed method achieves highly competitive performance compared to general video object removal methods.

**Strengths:**

The proposed solution for removing the video subtitles is straightforward and easy to implement. That would be very helpful if the authors release the constructed dataset.

**Weaknesses:**

1. The reviewer is a bit concerned about the robustness of mask-free subtitle removal. Although the authors proposed a data construction pipeline to generate various kinds of subtitles, this data may not cover all types of subtitles in real-world videos. Meanwhile, if the video contains other text (not subtitles), does the model also remove this text? The authors should conduct more experiments on videos with various types of text to test this.

2. The proposed model is straightforward, but I recommend the authors explore more designs for the reference video conditioning strategy and the long video inference strategy. Meanwhile, it seems that the prompt instruction in this setting is not necessary, so what would happen if the input prompt were removed? In addition, although the method is faster than existing diffusion-based video object removal models, it is still slow and far from real-time inference, which is important in practical application scenarios.

3. The evaluation is insufficient. The authors only compared their model to only one existing general video object removal model (Minimax-Remover). The related work section mentions several other relevant video inpainting and object removal methods (e.g., ProPainter, DiffuEraser, EraserDiT), none of which are included in the comparison. Moreover, the authors should also include results from classical (non-deep learning) methods to better demonstrate the superiority of the proposed method.

**Questions:**

See the weakness.

---

> ### Author Response · Authors · 2025-11-20
>
> 1. Our data construction process strives to include as many common subtitle styles as possible, enabling effective subtitle removal in most scenarios. While the current model cannot achieve 100% removal for every subtitle style, we have included failure cases in the "Limitations" subsection. The model demonstrates strong generalization capabilities—for instance, although Russian subtitles were not included during training, the model performs well on Russian content (as shown in Figure 6).
>
> Figure 14 in the Appendix illustrates the model's ability to remove subtitles while preserving other textual elements. A full video comparison is provided in the supplementary material (demo1.mp4).
>
> 2. The simplicity and effectiveness of the current model architecture reflect our original design intention. In future work, we plan to further explore more efficient conditional control strategies and long-video inference techniques.
>
> Since the subtitle removal task is trained with a fixed prompt, the impact of the fixed  user prompt is indeed limited. Please refer to Figure 12 in Appendix.
>
> However, the current framework is extensible to other video editing tasks—for example, the video stylization task included in the Appendix, where different styles require different prompts. We retained the user prompt branch to facilitate future extensions of the model to a broader range of video editing applications. Moreover, keeping a fixed user prompt during inference does not introduce additional interaction for subtitle removal and therefore does not affect the user experience.
>
> The current model still has room for further speed improvements. In the revised version of the paper, we present a comparison of inference performance using 4, 2, and 1 steps (Please see Figure 11 in the Appendix). One-step inference already represents the limit of step optimization for current diffusion models.
>
> 3. Since minimax-remover is currently the state-of-the-art model in the video subtitle removal domain and is based on the Video DiT framework, we primarily compare our method against it. EraserDiT has not been open-sourced yet. DiffuEraser, which is built on the SD1.5 diffusion model, can only handle videos with 720p resolution, whereas our method is based on Video DiT and supports higher resolutions, so we did not include it in the comparison.
>
> To provide  a more comprehensive demonstration of the superiority of the proposed framework, we have added comparative results of Propainter and DiffuEraser on the VSR-Bench-400 dataset. Although the optimal processing resolution for these two methods is not 1080p, we nevertheless enforced the models to process 1080p videos. All results are directly generated by the models without any post-processing. The comparative results are presented in the Table below.
>
> | Method     | PSNR ↑ | SSIM ↑ | LPIPS ↓ | VFID ↓ | Time ↓ |
> |------------|--------|--------|--------|--------|--------|
> | Propainter     | 26.8198  | 0.8230  | 0.1778 | 374.40 | 38s |
> | DiffuEraser   | 27.5084  | 0.8145  | 0.1222 | 303.05 | 166s |
> | Minimax-Remover (6-step) | 28.3109  | 0.8785  | 0.1011 | 252.70 | 150s |
> | SEDiT (1-step) |  31.5863 | 0.8805  | 0.0981 | 105.18 | 2s |
>
> It can be observed that our method demonstrates a significant advantage in terms of the FVD metric. In addition, we have supplemented visual comparisons in the appendix, as shown in Figures 15, 16, and 17.
>
> Traditional non-deep learning approaches still fall short in performance compared to diffusion-based methods, especially when dealing with complex subtitle scenarios commonly found in short-form social media videos. As illustrated in Figure 14 of the appendix, mask-based methods are limited when the mask is inaccurate.

---

> > ### Author Response · Authors · 2025-11-28
> >
> > Dear Reviewers,
> >
> > The manuscript has been revised in accordance with the most recent reviewer comments, and the modified sections are highlighted. We hope that our responses resolve your concerns, and kindly ask you to inform us if there are any further points that need to be addressed.

---

### Official Review · Reviewer_2h8Z · 2025-11-02

**Soundness:** 2
**Presentation:** 1
**Contribution:** 1
**Rating:** 2
**Confidence:** 4

**Summary:**

This paper proposes a DiT-based end-to-end video subtitle erasure network (SEDiT). To this end, the clean latent of the reference video and the noise latent of the target video are merged using a token-level concatenation in the sequence dimension. Then, they are conditioned by user-prompt subtitle erasing prompts. Additionally, new datasets for video subtitle erasure are introduced with a proposed data synthesis pipeline.

**Strengths:**

Belows are the strong points that this paper has:

1. The proposed architecture for video subtitle erasure is simple and straightforward, requiring no special modifications to the base model.

2.  As illustrated in Figure 3, the data synthesis pipeline is systematically designed, effectively considering the various characteristics and artifacts introduced during subtitle acquisition.

**Weaknesses:**

Belows are the weak points that this paper has:

1. mask M$_{subtitle}$ in Equation (2), which denotes the subtitle area, appears to contradict SEDiT’s claimed mask-free training approach. The authors should clarify this inconsistency.

2. The paper lacks sufficient discussion of related work and baseline comparisons. Subtitle erasure closely relates to video decaptioning, for which several prior studies exist (e.g., BVDNet [1], Deep Video Decaptioning [2]). The authors should provide a self-contained explanation of the decaptioning task and demonstrate the proposed model’s advantages over existing decaptioning methods.
[1] Kim et al., Deep Blind Video Decaptioning by Temporal Aggregation and Recurrence
[2] Chu et al., Deep Video Decaptioning

3. Although the proposed architecture is straightforward, it shows limited innovation specific to subtitle erasure. The method primarily combines an existing flow-based model with conditioning techniques, without offering substantial architectural contributions or theoretical insights expected for an ICLR-level paper.

4. The paper lacks ablation studies on key design choices, such as the 3D RoPE used for reference and noisy latents, the number of conditioning frames in the chunk-wise strategy, and the performance on long-video subtitle erasure.

5. While the authors emphasize the importance of efficient VAE modeling, details regarding the design and implementation of the “highly compressed VAE” are missing and should be elaborated.

**Questions:**

Please answer my questions listed above in Weaknesses.

---

> ### Author Response · Authors · 2025-11-20
>
> 1. We thank the reviewer for pointing this out. In the initial stage, the model was trained using only MSE loss. Later, we introduced a focal loss targeting the subtitle region for better optimization. Accordingly, we have updated the terminology from "mask-free training" to "mask-free inference."
>
> 2. We thank the reviewer for highlighting the two video decaptioning works. We have added descriptions of them in the related work section. Since both of these methods were trained on models with a resolution of $128 \times 128$, they cannot be directly applied to inference on 1080p videos. Therefore, we conducted comparative experiments to evaluate their performance. In terms of architecture, both prior works adopt generative adversarial networks, while our method is based on the DiT framework. In terms of performance, our approach achieves superior restoration of fine-grained textures, although the previous methods are faster.
>
> 3. The representative work VACE in the video editing domain adopts channel-wise conditional cascading for full-parameter fine-tuning, or introduces additional context blocks to encode conditions. In contrast, our current work draws inspiration from state-of-the-art instruction-based image editing methods, such as Flux-Kontext and Qwen-Image-Edit, and extends their ideas to the subtitle removal task in video editing. Furthermore, in the revised version of the paper, we demonstrate the effectiveness of our architecture in video stylization tasks, highlighting its generalizability and its potential contribution to advancing the field of video editing.
>
> 4. Regarding the ablation study questions:
>
> (1) The motivation behind our 3D RoPE design stems from the frame-wise alignment nature of the video subtitle removal task. Therefore, we use identical positional encoding for reference and noisy latents: $u_{\text{ref}} = u_{\text{noise}} = (f, h, w)$. In the LTX-Video model, the upper limit of $f$ is set to 20, corresponding to a video length of $20 \times 8+1=161$ frames. If we adopt a shifted encoding scheme, such as $u_{\text{ref}} = (f, h, w), u_{\text{noise}} = (f + \delta, h, w)$, then the maximum $f$ becomes 10, limiting the reference video to $10 \times 8 + 1 = 81$ frames. This means a single chunk can only process up to 81 frames, which is insufficient for our 121-frame setting in 720p resolution.
> Another alternative is to introduce an extra dimension to distinguish reference and noisy latents: $u_{\text{ref}} = (0, f, h, w), u_{\text{noise}} = (1, f, h, w)$. However, this 4D RoPE has a different feature dimensionality from the original 3D RoPE, requiring retraining of the LTX-Video base model and significantly increasing computational cost. Therefore, we chose the most direct, effective, and semantically clear approach.
>
> (2) Regarding the number of conditioning frames, we initially experimented with using a single frame and found it sufficient to resolve intra-shot discontinuities. We did not increase the number of conditioning frames further, as doing so would reduce the number of output frames per chunk and negatively impact overall video processing speed.
>
> (3) As for long-video subtitle removal performance, we have provided high-quality comparative videos in the supplementary material, which can be downloaded and viewed locally.
>
> 5. The goal of this paper is to develop a subtitle removal framework that can be applied at large-scale, where both processing speed and erasure quality are equally important. Our emphasis on the high compression rate of the VAE is intended to motivate the choice of LTX-Video as the base model. This enables our method to directly handle 1080p and higher-resolution videos with reasonable processing speed. The 3D VAE architecture itself is a contribution of the LTX-Video base model, which is why we do not elaborate on it further in this paper.

---

> > ### Author Response · Authors · 2025-11-28
> >
> > Dear Reviewers,
> >
> > The manuscript has been revised in accordance with the most recent reviewer comments, and the modified sections are highlighted. We hope that our responses resolve your concerns, and kindly ask you to inform us if there are any further points that need to be addressed.

---

### Official Review · Reviewer_Hfxz · 2025-11-03

**Soundness:** 4
**Presentation:** 4
**Contribution:** 4
**Rating:** 10
**Confidence:** 4

**Summary:**

Mask-free video editing object-removal approach (whole-video, not just a few frames) based on prompt. This is much simpler for the user, and reduce risks of artifacts from the mask. They apply their approach specifically to remove hardcoded subtitles. This could be especially useful in situation were an old fringe movies is only available through a torrent with hardcoded Russian subtitles. I would definitively use this. This would be very useful to archivists who tried to keep high-quality of old media alive. This is high impact.

They concatenate the reference video with the noisy target video. Text embedding is injected as cross-attention. They have the loss focus more on the subtitle region in order to not focus on reconstruction of the whole scene but specifically the subtitle region. They generate different font at different position and styles. It used chunk-wise of frames to ensure good quality. They found that a single-frame of overlap between chunks is good enough for satisfactory temporal consistency. Based on 400k videos from Pexel website. They make a benchmark from 200 videos. They use LoRA (19% trainable params)

This is amazing work: this is a simple solution to a simple but important problem. This is high-impact work.

12sec seems slow for 65 frames though. But its not that bad actually. A 30fps movie would take 6h to remove the subtitle. I would add that kind of number to the paper because its quite powerful. This means that with 4 gpus, you could do it in less than 2h.

It would be great to add some "worst cases" where we see artifacts because nothing is perfect, there must be some. Its interesting to see that minimax-remover has clear artifacts sometimes, but not your method.

Why not remove the user prompt? Since you are training for a single task this seems unnecessary to have the T5 model on the same prompt.

**Strengths:**

See "Summary"

**Weaknesses:**

See "Summary"

**Questions:**

See "Summary"

---

> ### Author Response · Authors · 2025-11-20
>
> We deeply appreciate the reviewers' recognition of our work and the valuable feedback they provided.
>
> (1) Regarding the inference speed issue:
>
> Under 6-step inference, the current SEDiT model takes approximately 12 seconds to process a 65-frame 1080p video. However, there is still room for improvement. I observed that reducing the number of inference steps has minimal impact on subtitle removal, which is a strongly reference-driven task. This observation has included in the revised version of the paper. Please refer to Figure 11 in the Appendix.
>
> (2) Regarding failure cases:
>
> We have added several observed failure cases to the revised paper. Minimax-Remover is a mask-based and general-purpose video inpainting model, which may be limited in subtitle removal performance due to the precision of the subtitle mask.
>
> (3) Regarding the user prompt:
>
> Removing the prompt-related components would require eliminating the text cross-attention module from the base model, which in turn demands large-scale retraining to achieve satisfactory results. Our intention is to preserve as much of the original model’s capabilities as possible. Moreover, the current architecture is extensible to other video editing tasks—we have added video stylization results in the revised paper to demonstrate this. In such scenarios, the user prompt plays a crucial role. Please see Figure 13 in Appendix.
>
> Taking all factors into account, we decided to retain the prompt branch.

---

> > ### Comment · Reviewer_Hfxz · 2025-11-20
> >
> > Thank you.
> >
> > The paper is better having now more interesting examples and video style transfer examples.
> >
> > I agree with other reviewers that comparing to more baselines would benefit the paper, but its not essential in my pov considering that they already compare to SOTA and the second-best, DiffuEraser, is 720p as the authors mention. Meanwhile, EraserDiT has not been open-sourced.
> >
> > I really think that this is a strong and very useful method. I might appear too positive with my 10/10, but I see value in it and I think that other reviewers are too harsh for such a simple and good solution to a high impact problem. I will keep my score.

---

### Author Response · Authors · 2025-11-30
**Summary of Rebuttal**

We sincerely thank the ACs and the Reviewers for their efforts on this matter.

$\textbf{Paper Summary}$

In this work, we introduces $\textbf{SEDiT}$, a mask-free framework for video subtitle removal. The key contributions are outlined as follows:

$\textbf{1. Mask-free subtitle erasure.}$ We propose a novel framework that eliminates the need for explicit subtitle detection and mask generation. This design avoids the performance degradation caused by subtitle mask errors and simplifies the overall pipeline. As illustrated in Fig. 14 of the Appendix, OCR-based approaches may suffer from false detections (e.g., failing to distinguish subtitles from non-subtitle text), whereas our method effectively addresses such cases. Furthermore, we construct a high-quality synthetic data pipeline encompassing font attributes, subtitle rendering directions, and transition effects. Leveraging this comprehensive synthetic dataset, our method achieves highly satisfactory subtitle removal results.

$\textbf{2. Efficient training with preserved generative capacity.}$ Inspired by state-of-the-art (SOTA) image editing techniques, we extend them to the video domain by introducing reference sequences through cascaded integration along the temporal dimension. This design reduces training cost while retaining the generative ability of the base model. Moreover, the framework can be readily adapted to other video editing tasks. As a demonstration, we present a video stylization case study (Fig. 13), which highlights the potential applicability of our approach to broader video editing scenarios.

$\textbf{3. Scalability to long videos.}$ Since subtitle removal is essentially a strong reference generation problem, we find that employing first-frame overlap yields satisfactory temporal consistency. For instance, processing a 1080p video with 65 frames requires only ~2 seconds of denoising time with the diffusion model, which demonstrates the feasibility of large-scale deployment.

$\textbf{Main Revision Notes:}$

$\textbf{1. Related Work}$

Responses to Reviewer 2h8Z

We have added two GAN-based works on Video Decaptioning in the Related Work section. These methods were trained on video data with a resolution of 128×128, making them difficult to directly apply to 1080p subtitle removal. Moreover, our approach is based on DiT, so we did not include experimental comparisons with these GAN-based methods.

$\textbf{2. Experimental Section}$

Responses to Reviewer Hfxz, Reviewer 5uKc, Reviewer Hrxk

We reconstructed the video subtitle removal benchmark dataset VSR-Bench-400 and compared our method against current mask-based approaches, including Propainter, DiffuEraser, and Minimax-Remover. Real subtitle masks were used to ensure fairness in comparison.

We further examined the effect of inference steps on performance and found that, for this strong reference task, even a single inference step can yield satisfactory results. This finding supports the scalability of our approach for large-scale applications.

Additionally, we added a failure case section to show the limitations of our model. For very short clips (<10 frames), incomplete erasure may occur (Fig. 10(a)). When severe blur transition effects are present, subtitles cannot be removed; however, normal subtitles, even those with very large fonts, can be effectively handled (Fig. 10(b)).

$\textbf{3. Fixed Prompt Setting: }$

Responses to Reviewer Hfxz, Reviewer 5uKc, Reviewer Hrxk

The current fixed prompt served only as a placeholder. We supplemented results under different prompts (Fig. 12) and found that, for the present fixed-prompt training, prompts have little impact during testing.

Consequently, we revised the paper title from “SEDiT: Mask-Free Video Subtitle Erasure with Prompt Instruction” to “SEDiT: Mask-Free Video Subtitle Erasure via Diffusion Transformer.” However, in our extended experiments on video stylization, different prompts can guide the model to generate distinct styles.

---

### Meta-Review · Area_Chair_TDf3 · 2025-12-23

**Summary:**

This paper introduces a mask-free inference approach for Video Subtitle Erasure. In the first round, this paper received four reviews (10 4 2 2). The two reviewers both challenged the model's innovation, arguing that the approach combines existing flow-based models with conditioning techniques, and its core contribution lies in adaptation or practical engineering design rather than a novel method. Although this paper has its merits, it was rejected due to the majority of negative feedback.

**Reviewer Concerns:**

In their rebuttal, the authors included comparisons with methods such as Minimax-Remover and DiffuEraser, while clarifying any inappropriate expressions. However, I also worry that their approach is an extension of SOTA instruction editing methods to subtitle removal rather than a novel method.

**Reviewer Scores:**

Even after full discussion, concerns about the novelty of the approach remain unresolved.

---

### Decision · Program_Chairs · 2026-01-26

Reject